# Synthesis, Antimicrobial and Mutagenic Activity of a New Class of d-Xylopyranosides

**DOI:** 10.3390/antibiotics12050888

**Published:** 2023-05-10

**Authors:** Karol Sikora, Piotr Szweda, Karolina Słoczyńska, Justyna Samaszko-Fiertek, Janusz Madaj, Beata Liberek, Elżbieta Pękala, Barbara Dmochowska

**Affiliations:** 1Faculty of Chemistry, University of Gdańsk, Wita Stwosza 63, 80-308 Gdańsk, Poland; karol.sikora@gumed.edu.pl (K.S.); j.samaszko-fiertek@ug.edu.pl (J.S.-F.); janusz.madaj@ug.edu.pl (J.M.); beata.liberek@ug.edu.pl (B.L.); 2Department of Inorganic Chemistry, Faculty of Pharmacy, Medical University of Gdańsk, Al. Gen. J. Hallera 107, 80-416 Gdańsk, Poland; 3Department of Pharmaceutical Technology and Biochemistry, Faculty of Chemistry, Gdańsk University of Technology, G. Narutowicza 11/12, 80-233 Gdańsk, Poland; piotr.szweda@pg.edu.pl; 4Department of Pharmaceutical Biochemistry, Faculty of Pharmacy, Jagiellonian University Medical College, Medyczna 9, 30-688 Krakow, Poland; karolina.sloczynska@uj.edu.pl (K.S.); elzbieta.pekala@uj.edu.pl (E.P.)

**Keywords:** d-xylosides, quaternary ammonium salts, NMR, antimicrobial activity, mutagenicity

## Abstract

Eight *N*-[2-(2′,3′,4′-tri-*O*-acetyl-α/β-d-xylopyranosyloxy)ethyl]ammonium bromides, a new class of d-xylopyranosides containing a quaternary ammonium aglycone, were obtained. Their complete structure was confirmed using NMR spectroscopy (^1^H, ^13^C, COSY and HSQC) and high-resolution mass spectrometry (HRMS). An antimicrobial activity against fungi (*Candida albicans*, *Candida glabrata*) and bacteria (*Staphylococcus aureus*, *Escherichia coli*) and a mutagenic Ames test with *Salmonella typhimurium* TA 98 strain were performed for the obtained compounds. The greatest activity against the tested microorganisms was shown by glycosides with the longest (octyl) hydrocarbon chain in ammonium salt. None of the tested compounds exhibited mutagenic activity in the Ames test.

## 1. Introduction

Taking into account the ease of obtaining from natural sources and the biological activity of its derivatives, d-xylose, next to d-glucose, should probably be considered the most important monosaccharide. Due to the fact that it can be obtained using hydrolysis of hemicellulose, constituting up to 30% of this sugar [1,2], d-xylose is the second—after d-glucose—most common monosaccharide found in nature. This puts it in the group of extremely important raw materials for obtaining chemicals and biofuels. An example is the biochemical synthesis of ethanol. The method of glucose (hexose) fermentation is widely known for its preparation. However, it turns out that equally good results are obtained using d-xylose or L-arabinose (pentoses) [3,4,5,6]. Another example of industrial use of hemicellulose—and, indirectly, d-xylose—is the production of d-xylitol by bacteria that produce enzymes necessary for d-xylose metabolism [7]. Due to the lower calorific value in relation to d-glucose (2.4 cal/g compared to 4.0 cal/g) and, above all, insulin-independent metabolism, xylose is commonly used as a sweetener. Discovered in 1891, it was used during World War II as a replacement for sucrose. Since then, a lot has been written about the benefits of its use and it is easy to find examples in the literature [8].

However, xylose is not only of industrial importance. All chiral atoms in d-xylopyranose have the same configuration as in d-glucopyranose; the only difference is the lack of one chiral center resulting from the lack of a terminal hydroxymethyl group. Perhaps, this similarity results in the high activity of naturally occurring d-xylopyranosides and this makes d-xylopyranosides good model compounds for comparative studies with d-glucopyranosides. Many examples of d-xylopyranosides exhibiting interesting biological properties have been described in the literature [9,10]. Terpene xylosides (Figure 1A–D) are commonly found in various herbs used in folk medicine, which, in addition to such properties for use as an anti-inflammatory or their ability to treat hot flashes during menopause, are also credited with the ability to treat Alzheimer’s disease.

Another type of naturally occurring d-xylopyranosides are derivatives containing polycyclic aromatic compounds such as aglycone [15,16,17]. Quercetin, a natural flavonol, is among the most common aglycones of this type (Figure 1E).

Another example of a naturally occurring d-xylopyranoside containing a polycyclic aromatic compound such as an aglycone is 4-*O*-(2′,3′,4′-tri-*O*-methyl-β-d-xylopyranosyl) diphyllin (Figure 1F).

Studies on this compound and its analogues conducted on several cell lines show that they may have anticancer properties [18].

Among the d-xylopyranosides that are worth mentioning is *p*-nitrophenyl-β-d-xylopyranoside (Figure 1G). Studies have shown that it can significantly affect the synthesis of proteoglycans and glycosaminoglycans [19,20].

Ellipticine is an alkaloid commonly found in plants of the Apocynaceae family. As has been shown, it exhibits antitumor activity. However, due to the hydrophobic nature of this polycyclic compound, most of its derivatives are sparingly soluble in water, which makes their use in medicine quite limited.

Honda et al. [21] synthesized a number of ellipticine derivatives in which they attached sugar derivatives, including d- and l-xylopyranose and xylofuranose (Figure 2). This was intended, among other things, to improve the solubility of the obtained derivatives. It is also worth noting that the aglycone in the obtained compounds was a quaternary ammonium salt. The obtained results of biological tests showed that two of the tested compounds, including the β-d-xylofuranose derivative, turned out to be very interesting and were to be submitted for further research.

However, in addition to d-xylopyranosides with polycyclic or aromatic aglycones, those with an aliphatic aglycone are also worth mentioning. Such xylopyranosides were reported by Wu et al. [22]. They described the synthesis of d-xylopyranosides with an aglycone containing two to four repeating oxyethyl (-OCH_2_CH_2_-) fragments (Figure 3). They used a tetra-*O*-acetyl derivative of d-xylose as a substrate for the preparation of glycosides. For the obtained compounds, they conducted water solubility and surface activity tests, stating a clear influence of the length of the alkyl chain on the investigated properties.

Studies on cationic surfactants have shown that they have a number of interesting biological activities. Quagliotto et al. [23] synthesized a new class of surfactants known as glucocationic compounds (Figure 4). These compounds, similar to those described above and prepared by Wu, contain a long-chain aglycone and exhibit surface activity. However, unlike the previous ones, their aglycone is a quaternary ammonium salt in which the nitrogen atom is connected to the anomeric carbon atom of the sugar moiety with an oxyethyl linker. As a substrate for the quaternization reaction, they used 2-bromoethyl-2,3,4,6-tetra-*O*-acetyl-β-d-glucopyranoside.

In this paper, we present the results of our research on the synthesis and biological activity of a new group of d-xylopyranosides with an aglycone possessing a quaternary ammonium salt. The presented group of xylopyranosides constitutes a group of xylocationic compounds. The aim of this work was to continue research on this group of glycocationic glycosides and to determine the effect of the lack of a terminal hydroxymethyl group in d-xylopyranosides on their biological activity compared to previously synthesized and tested glycosides [24,25] with the d-*gluco* and d-*galacto* configuration, of which only the latter showed weak mutagenicity in the Ames test.

## 2. Results and Discussion

### 2.1. Chemistry

As a result of the classical acetylation of d-xylose, 1,2,3,4-tetra-*O*-acetyl-d-xylopyranose (**1**) was obtained as a mixture of α and β anomers. This mixture was used without further purification for the glycosylation reaction with 2-bromoethanol in the presence of BF_3_·Et_2_O (Figure 1).

Contrary to the syntheses of the analogous glycosides of hexopyranose derivatives previously reported by us [23], it was not possible to select the reaction conditions so that only single anomer of the xylopyranoside was formed. Even though Dahmen et al. [26] reported that, in the case of per-*O*-acetylated d-xylopyranose, they were able to obtain only one anomer of the glycoside, which they explained by the formation of the thermodynamically more stable 1,2-*trans* glycoside, we were unable to obtain analogous results. Regardless of the reaction time used and the proportion of substrates, each time, a mixture of two products (α and β anomers) was obtained in a similar quantitative ratio. In the end, a synthesis time of 1 h on ice and then 8.5 h at room temperature turned out to be the optimal conditions. Anomers as isomers show large similarity in structure, which results in very similar properties and thus the difficulty in separating them. The use of chromatography for their separation leads to three fractions: two were pure anomers and the third was their mixture. This means that pure compounds are obtained in relatively low yields. In our case, chromatographic separation using flash chromatography in the *n*-hexane:ethyl acetate (3:1), using a column packed with silica gel, allowed us to isolate 2-bromoethyl 2′,3′,4′-tri-*O*-acetyl-β-d-xylopyranoside (**2**) in 10% yield and 2-bromoethyl 2′,3′,4′-tri-*O*-acetyl-α-d-xylopyranoside (**3**) in 7% yield.

The structure of compounds **2** and **3** was confirmed based on NMR spectroscopy (chemical shifts and coupling constants are included in the experimental section). The *J*_1,2_ coupling constant of 6.4 Hz indicates the *trans* proton configuration, which confirms the β configuration of compound **2**, while the value of 3.6 Hz in compound **3** corresponds to the *cis* proton arrangement as in the α anomer. In the spectra, apart from the characteristic signals of the sugar ring protons, are signals of nine protons of methyl groups from three *O*-acetyl groups. The spectra also show the protons of the two methylene groups (3.5–4.1 ppm) of the aglycon. The signals *m*/*z* 405.1 [M + Na]^+^ appearing in the MALDI-TOF MS spectra additionally confirmed the structure of compounds **2** and **3**.

2-Bromoethyl 2′,3′,4′-tri-*O*-acetyl-β-d-xylopyranoside (**2**) was used as a substrate in the quaternization reaction with selected amines (pyridine, trimethylamine, *N*,*N*-dimethylhexylamine and *N*,*N*-dimethyloctylamine). In order to determine the optimal reaction conditions, different solvents and different temperatures were tested. TLC chromatography and MALDI-TOF MS analysis were used to control the progress of the reaction. Ultimately, it turned out that for all the amines used, the best yields were obtained at 70 °C. In most cases, except for trimethylamine, the optimal reaction time was 24 h. Detailed data on the reaction conditions are provided in Appendix A).

The synthesis of the pyridine derivative **4a** did not require the use of an additional solvent as it was both a substrate and a solvent. In the case of trimethylamine, the reaction was carried out in a 30% ethanol solution of the amine. The analysis of the obtained products showed that under the reaction conditions, apart from the quaternization reaction, an undesirable partial de-*O*-acetylation of the resulting product took place. Therefore, in order to obtain the expected product **4b**, all volatile components were removed from the post-reaction mixture under reduced pressure, and the residue was subjected to exhaustive *O*-acetylation. When *N*,*N*-dimethylhexylamine and *N*,*N*-dimethyloctylamine were used for the quaternization reaction, compounds **4c** and **4d** were obtained with similar yields. In this case, the reaction required the use of acetonitrile as a solvent. Its selection was made on the basis of a series of test reactions. Acetonitrile allowed both easy observation of the reaction progress (using TLC) and ensured good solubility of the substrates. The high boiling point of the substrate amines made their removal by evaporation under reduced pressure difficult and required extraction with diethyl ether and *n*-hexane after dissolving the reaction mixture in water.

^1^H and ^13^C NMR spectra were recorded to confirm the structure of the obtained salts **4a–d**. The signal assignments were made based on the analysis of COSY and HSQC spectra. In all spectra, the H-1 proton signals appear in the form of a doublet with the *J*_1,2_ coupling constant with values ranging from 5.9 to 6.8 Hz, confirming the β configuration of the anomeric center. In addition to the anomeric proton signal, signals from the protons of the sugar ring are present in the spectra. The analysis of their coupling constants reveals that the pyranose ring in β-d-xylopyranosides **4a–d** is in the conformational equilibrium between the ^4^C_1_ and ^1^C_4_ conformations, shifted towards the ^4^C_1_ conformation. This conclusion is supported by the *J*_2,3_ 7.2–8.8 Hz, *J*_3,4_ 7.5–8.4 Hz and *J*_4,5′_ 8.0–8.4 Hz coupling constants, which indicate that the respective protons are not fully axially oriented. The ethyl linkage connecting the sugar unit and the quaternary nitrogen atom occurs in the form of three groups of signals. The protons of the methylene group bounded to the oxygen atom of the sugar part appear as two signals at about 4.00 ppm and 4.25 ppm. The protons of the second methylene group of the linker directly linked to the quaternary nitrogen atom of the pyridinium salt **4a** appear as a signal at 4.78 ppm. This value of the chemical shift is due to the fact that the nitrogen atom of this salt is present in the aromatic (pyridine) ring. In the case of the aliphatic amine salts **4b–d**, the signal of these protons occurs at about 3.58 ppm. The signals from the aromatic pyridine ring appear as two doublets and a triplet at 8.01, 9.49 and 8.46 ppm, respectively. In the aliphatic amines salts, the methyl groups bound directly to the nitrogen atom occur at about 3.06–3.12 ppm. The proton signals of the methylene groups of the hydrocarbon chain (in compounds **4c** and **4d**) were observed at the 3.30, 1.72 and 1.28 ppm, where the first value is for the methylene group directly bonded to the nitrogen atom. Signal of the terminal methyl group is recorded at 0.83 ppm. A complete description of the NMR spectra signals is provided in the experimental section.

The synthesis of α-d-xylopyranosides **5a–d** was performed analogously to that described above. The yields of the quaternization reaction (about 70%) were similar to those obtained for d-xylopyranoside with β configuration of the anomeric carbon atom. The exception is compound **5b**, obtained by reaction with trimethylamine, which was obtained in a clearly lower (55%) yield (see Appendix A). Moreover, for these products, their structure was confirmed using NMR spectroscopy and mass spectrometry.

The H-1 proton signals in the ^1^H NMR spectra of **5b–d** appear as doublets with a *J*_1,2_ coupling constant of 2.8–3.3 Hz, which confirms the α configuration of the anomeric center. The exception was the signal of the H-1 proton of salt **5a**, which was in the form of a broad singlet. The values of 9.2–9.4 Hz of the remaining coupling constants (*J*_2,3_, *J*_3,4_ and *J*_4,5_) of the protons present in the sugar ring indicated the ^4^C_1_ conformation of the sugar ring of the d-xylopyranosides with α configuration. Proton signals from the aglycone part of **5a–d** are placed similarly to those recorded for **4a–d**.

All new synthesized *N*-[2-(2′,3′,4′-tri-*O*-acetyl-α/β-d-xylopyranosyloxy)ethyl]-ammonium bromides are water soluble.

### 2.2. Antimicrobial Activity

All synthesized salts **4a–d** and **5a–d** were tested for their antimicrobial activity against fungi (*Candida albicans*, *Candida glabrata*) and bacteria (*Staphylococcus aureus*, *Escherichia coli*). The minimum inhibitory concentration (MIC, μg/mL) was determined for the tested strains. Table 1 summarizes the MIC values of the studied compounds.

The data in Table 1 indicates that d-xylopyranosides **4c**, **4d** and **5d** show the ability to inhibit the growth of both fungi and bacteria. These compounds are characterized by the presence of a relatively long alkyl chain on a quaternary nitrogen atom. Among them, compound **4c**, with a hexyl group, in addition to two methyl groups, possesses the lowest activity. Two remaining glycosides, **4d** and **5d**, having an octyl group, in addition to two methyl groups, show greater activity in the conducted studies. Two more carbon atoms in the chain cause the molecule to be more hydrophobic. Glycosides **4d** and **5d** are isomers that differ solely in the configuration of the anomeric carbon atom of the sugar moiety. The obtained results indicate twofold higher activity of the compound **5d** (α anomer) in three out of four tested microorganisms. We observed a different situation in the case of the shorter hexyl substituent. While the compound **4c** (β anomer) shows some slight activity against most of the tested microorganisms, its α isomer (**4d**) shows no activity. On the basis of these results, it is possible to unequivocally indicate the significant influence of the hydrophobic nature of the substituents on the activity of the tested compounds.

Glycoconjugates, i.e., compounds in which sugar fragments are bonded to non-sugar ones, can penetrate the cell membrane in two ways. The first one is based on the active transport through the GLUT membrane proteins. The second mechanism is passive migration (diffusion of a lipophilic compound through biological membranes). Transport using GLUT proteins is more preferred for compounds with free hydroxyl groups, as is the case with intracellular glucose transport. It can be influenced by many factors, such as the structure of the sugar compound, the type of its substitution or even the type of aglycone [27,28]. In addition, studies indicate that, due to better binding to GLUT transport proteins, this mechanism prefers compounds with the β configuration of the anomeric carbon atom [29]. The passive mechanism is preferred in the case of sugar compounds containing substituted hydroxyl groups and a hydrophobic aglycone. The obtained results of biological tests seem to indicate that the tested d-xylopyranosides overcome the barrier of the cell membrane based on the mechanism of diffusion and this seems to be the right direction when designing further syntheses.

Furthermore, xylopyranosides **4d** and **5d**, which showed good activity against the reference strains, were tested for antistaphylococcal activity against clinical isolates collected from patients with skin and soft tissue infections as well as strains derived from subclinical bovine mastitis milk sample infections. In this study, 20 strains were used, and the MIC values are summarized in Table 2.

Considering clinical isolates, both tested compounds showed activity similar to that obtained against the reference strain of *S. aureus*. Compound **5d** exhibits two to four times higher activity and no differences in susceptibility were observed for MRSA (methicillin-resistant *Staphylococcus aureus*) and MSSA (methicillin-sensitive *Staphylococcus aureus*) isolates.

In addition to activity against selected fungi and bacteria, all obtained d-xylopyranosides were tested for mutagenic activity using the Ames test. This test is commonly used and was recommended by OECD to detect the mutagenic potential of chemicals [30]. The results of the analyses are summarized in Table 3. In the Ames test, compounds in which a dose–response relationship is detected and an at least twofold increase in the number of mutants (MI ≥ 2) is observed with at least one concentration are considered to have mutagenic activity.

As the test results showed, none of the studied compounds show mutagenic activity in this test.

## 3. Summary

The obtained *N*-[2-(2′,3′,4′-tri-*O*-acetyl-d-xylopyranosyloxyethyl)]ammonium bromides (**4a–d** and **5a–d**) were subjected to biological tests. They showed no mutagenic activity in the Ames test. On the other hand, the **4d** and **5d** salts, characterized by the presence of an octyl substituent on the nitrogen atom, showed antimicrobial activity against fungi (*C. albicans*, *C. glabrata*) and bacteria (*S. aureus*, *E. coli*) with an MIC value within the range of 32−256 μg/mL. These d-xylopyranosides also showed similar activity against clinical isolates of the *S. aureus* strain. Of these two stereoisomers, the compound with the α configuration of the anomeric center (**5d**) was more active.

## 4. Experimental Section

### 4.1. Synthetic Procedures and Results of Analysis

**2-Bromoethyl 2,3,4-tri-*O*-acetyl-β-d-xylo-** (**2**) **and α-d-xylopyranosides** (**3**)—2-bromoethanol (2.0 mL, 28 mmol) was added to a solution of compound **1** (1 g, 3.1 mmol) in dry CH_2_Cl_2_ (10 mL). The flask was cooled in an ice bath and boron trifluoride etherate solution (3.64 mL, 29.48 mmol) was slowly added dropwise. The reaction mixture was stirred for 1 h in ice and next for 8.5 h at room temperature. The reaction mixture was diluted with CH_2_Cl_2_ (10 mL) and washed with NaHCO_3_ aqueous solution (2 × 10 mL), water (2 × 10 mL), saturated aqueous NaCl solution (10 mL) and water (10 mL). The organic layer was dried over anhydrous MgSO_4_, filtered and concentrated under reduced pressure. Column chromatography (ethyl acetate-hexane 1:3) of the crude product first yielded compound **3** (85 mg, 7%, yellowish oil); R_f_ 0.25 (ethyl acetate-hexane 1:3); ^1^H-NMR (400 MHz, CDCl_3_): δ 5.48 (t, 1H, H-3, *J*_3,4_ 10.0 Hz); 5.08 (d, 1H, H-1, *J*_1,2_ 3.6 Hz); 4.96 (dt, 1H, H-4, *J*_4,5′_ 10.0 Hz, *J*_4,5_ 4.4 Hz); 4.79 (dd, 1H, H-2, *J*_2,3_ 10.0 Hz); 3.99 (dt, 1H, OCH_2a_, *J* 11.2 Hz); 3.78 (m, 2H, H-5, OCH_2b_, *J* 6.0 Hz); 3.69 (dd, 1H, H-5′, *J*_5,5′_ 10.8 Hz); 3.49 (t, 2H, CH_2_Br, *J* 6.4 Hz); 2.07–2.03 (3s, 9H, 3 × OAc); ^13^C NMR (100 MHz, CDCl_3_): 170.5–170.2 (3C, 3 × OO*C*CH_3_ ), 95.2 (C-1), 71.2 (C-2), 69.7 (C-3), 68.6 (2C, C-4 and O-CH_2_), 58.9 (C-5), 30.6 (CH_2_Br), 21.0–20.9 (3C, 3 × COO*C*H_3_); MALDI TOF- MS (DHB): *m*/*z* 405.1; 407.1 ([M + Na]^+^) and 421.1; 423.1 ([M + K]^+^).

Eluted second was **2** (121 mg, 10% yield, mp 127–129 °C,); *R*_f_ = 0.17 (ethyl acetate-hexane 1:3); ^1^H-NMR (400 MHz, CDCl_3_): δ 5.16 (t, 1H, H-3, *J*_3,4_ 8.4 Hz); 4.94 (m, 2H, H-2, H-4, *J*_2,3_ 8.4 Hz, *J*_4,5′_ 8.4 Hz); 4.55 (d, 1H, H-1, *J*_1,2_ 6.4 Hz); 4.14 (dd, 1H, H-5, *J*_4,5_ 5.2 Hz); 4.08 (dt, 1H, OCH_2a_, *J* 11.2 Hz); 3.48 (dt, 1H, OCH_2b_, *J* 6.4 Hz); 3.39 (t, 2H, CH_2_Br, *J* 6.0 Hz); 3.38 (dd, 1H, H-5′, *J*_5,5′_ 12.0 Hz); 2.07–2.04 (3s, 9H, 3 × OAc); ^13^C NMR (100 MHz, CDCl_3_): 170.2–169.7 (3C, 3 × OO*C*CH_3_ ), 100.9 (C-1), 71.3 (C-3), 70.6 (C-4), 69.4 (O-CH_2_), 69.0 (C-2), 62.3 (C-5), 30.2 (CH_2_Br), 21.0–20.9 (3C, 3 x COO*C*H_3_); MALDI TOF-MS (DHB): *m*/*z* 405.1; 407.1 ([M + Na]^+^) and 421.1; 423.1 ([M + K]^+^).

### 4.2. General Procedure for Quaternization Reaction

Amines (pyridine, 33% trimethylamine in ethanol, *N*,*N*-dimethylhexylamine or *N*,*N*-dimethyloctylamine in acetonitrile) and compounds **2** and **3**, respectively, were placed in a glass screw-capped ampoule. The mixture was heated at 70 °C for a certain period of time. Next, volatile compounds were evaporated under reduced pressure and the residue, after being dissolved in water, was extracted with chloroform (diethyl ether and then with *n*-hexane for *N*,*N*-dimethylhexylamine or *N*,*N*-dimethyloctylamine) three times. After concentration, thick syrup was obtained. Finally, product (*R*_f_ = 0.0 (ethyl acetate-hexane 1:3)) was dried in a vacuum desiccator over P_2_O_5_.

***N*-[2-(2′,3′,4′-tri-O-acetyl-β-d-xylopyranosyloxy)ethyl]pyridinium bromide (4a)**—A reaction between compound **2** (30.7 mg, 0.080 mmol) and dry pyridine (0.5 mL), carried out for 24 h, yielded compound 4a (25 mg, 67%, colorless oil); [$\alpha ]\begin{array}{c} 20\\ D \end{array}$−37.6° (c 1, MeOH); ^1^H NMR (500 MHz, D_2_O): δ 9.49, 8.46, 8.01, (d, t, t, 5 H, Py, J 5.6, J 7.2); 5.12 (t, 1H, H-3, J_3,4_ 8.4 Hz); 4.88 (m, 1H, H-4, J_4,5′_ 8.4 Hz); 4.78 (m, 2H, H-2, CH_2_-N, J_2,3_ 8.8 Hz); 4.67 (d, 1H, H-1, J_1,2_ 6.8 Hz); 4.26 and 4.09 (2 x m, 2H, O-CH_2a_ and O-CH_2b_, J 13.5 Hz J 6.0 Hz); 3.94 (dd, 1H, H-5, J_4,5_ 4.8 Hz); 3.46 (dd, 1H, H-5′, J_5,5′_ 12.4 Hz); 2.01–1.98 (3s, 9H, 3 × OAc); ^13^C NMR (125 MHz, D_2_O): 173.2–172.6 (3C, 3 × OOCCH_3_), 146.5–145.1, 128.3 (5C, Py), 99.8 (C-1), 71.5 (C-3), 70.8 (C-2), 68.8 (C-4), 67.6 (O-CH_2_), 61.5 (C-5), 61.3 (CH_2_-N), 20.4–20.3 (3C, COOCH_3_); MALDI TOF-MS (CHCA): *m*/*z* 382.2 ([M-Br]^+^), HRMS (ESI): *m*/*z* ([M-Br]^+^) calculated for C_18_H_24_O_8_N: 382.1496; found: 382.1492.

***N*-[2-(2′,3′,4′-tri-O-acetyl-β-d-xylopyranosyloxy)ethyl]-N,N,N-trimethylammonium bromide (4b)**—A reaction between compound **2** (37 mg, 0.097 mmol) with 33% trimethylamine in ethanol (0.5 mL) ended after 3 h. After the removal of volatile compounds, the residue was O-acetylated with Ac_2_O (0.5 mL) and pyridine (0.5 mL) at room temperature for 24 h. Purification of the crude product, as described in the general procedure, yielded compound 4b (36.9 mg, 87%, colorless oil); [$\alpha ]\begin{array}{c} 20\\ D \end{array}\;$−40.0° (c 1, MeOH); ^1^H NMR (500 MHz, D_2_O): δ 5.17 (t, 1H, H-3, J_3,4_ 7.6 Hz); 4.95 (td, 1H, H-4, J_4,5′_ 8.0 Hz); 4.89 (dd, 1H, H-2, J_2,3_ 7.6 Hz); 4.78 (d, 1H, H-1, J_1,2_ 6.0 Hz); 4.25 (d, 1H, O-CH_2a_, J 13.0 Hz); 4.13 (dd, 1H, H-5, J_4,5_ 4.4 Hz); 4.02 (d, 1H, O-CH_2b_); 3.58–3.55 (m, 3H, H-5′, CH_2_-N, J_5′,5_ 12.8 Hz); 3.12 (s, 9H, 3 × CH_3_ of N(CH_3_)_3_); 2.08, 2.05, 2.04 (3s, 9H, 3 × OAc); ^13^C NMR (125 MHz, D_2_O): 173.0, 172.9, 172.6 (3C, 3 × OOCCH_3_), 99.3 (C-1), 70.7 (C-3), 70.3 (C-2), 68.4 (C-4), 65.5 (CH_2_-N), 62.9 (O-CH_2_), 61.0 (C-5), 54.0, 54.0, 53.9 (3C, N(CH_3_)_3_), 20.2, 20.1 (3C, COOCH_3_); MALDI TOF-MS (CHCA): *m*/*z* 362.3 ([M-Br]^+^); HRMS (ESI): *m*/*z* ([M-Br]^+^) calculated for C_18_H_28_O_8_N: 362.1809; found: 362.1805.

***N*-[2-(2′,3′,4′-tri-O-acetyl-β-d-xylopyranosyloxy)ethyl]-*N*-hexyl-*N*,*N*-dimethylammonium bromide (4c)**—A reaction between compound **2** (35 mg, 0.092 mmol) and N,N-dimethylhexylamine (0.2 mL, 1.15 mmol) in acetonitrile (1.0 mL) carried out for 24 h led to compound 4c (34 mg, 72%, colorless oil); [$\alpha ]\begin{array}{c} 20\\ D \end{array}\;$−23.2° (c 1, MeOH); ^1^H NMR (500 MHz, D_2_O): δ 5.19 (t, 1H, H-3, J_3,4_ 7.7 Hz); 4.96 (m, 1H, H-4, J_4,5_ 4.4 Hz); 4.88 (t, 1H, H-2, J_2,3_ 7.2 Hz); 4.79 (d, 1H, H-1, J_1,2_ 5.9 Hz); 4.21 (d, 1H, O-CH_2a_, J 13.6 Hz); 4.14 (dd, 1H, H-5, J_4,5′_ 8.4 Hz); 4.00 (d, 1H, O-CH_2b_); 3.57 (m, 3H, H-5′, CH_2_N); 3.29 (t, 2H, N-CH_2_); 3.06 (s, 6H, 2 × CH_3_ of N(CH_3_)_2_); 2.08, 2.05, 2.04 (3s, 9H, 3 × OAc); 1.71 (bs, 2H, CH_2_); 1.28 (bs, 6H, 3 x CH_2_); 0.83 (bs, 3H, CH_3_); ^13^C NMR (125 MHz, D_2_O): 173.0, 172.9, 172.6 (3C, 3 × OOCCH_3_), 99.4 (C-1), 70.9 (C-3), 70.5 (C-2), 68.4 (C-4), 65.5 (N-CH_2_), 62.9 (O-CH_2_), 62.8 (CH_2_-N), 61.1 (C-5), 51.5 (2C, N(CH_3_)_2_), 30.5, 25.3, 21.9, 21.8 (4C, 4 x CH_2_), 13.2 (CH_3_), 20.3, 20.2 (3C, COOCH_3_); MALDI TOF-MS (CHCA): *m*/*z* 432.4 ([M-Br]^+^); HRMS (ESI): *m*/*z* ([M-Br]^+^) calculated for C_21_H_38_O_8_N: 432.2592; found: 432.2587.

***N*-[2-(2′,3′,4′-tri-O-acetyl-β-d-xylopyranosyloxy)ethyl]-*N*,*N*-dimethyl-*N*-octylammonium bromide (4d)**—A reaction between compound **2** (35 mg, 0.092 mmol) and N,N-dimethyloctylamine (0.2 mL, 0.972 mmol) in acetonitrile (1.0 mL) carried out for 24 h yielded 4d (35 mg, 70%, yellowish oil); [$\alpha ]\begin{array}{c} 20\\ D \end{array}\;$−19.4° (c 1, MeOH); ^1^H NMR (500 MHz, D_2_O): δ 5.19 (t, 1H, H-3, J_3,4_ 7.9 Hz); 4.96 (m, 1H, H-4); 4.88 (td, 1H, H-2, J_2,3_ 7.9 Hz); 4.79 (d, 1H, H-1, J_1,2_ 6.3 Hz); 4.22 (bd, 1H, O-CH_2a_); 4.14 (dd, 1H, H-5, J_4,5_ 4.6 Hz, J_5,5′_ 12.2 Hz); 4.01 (bd, 1H, O-CH_2b_); 3.58 (m, 3H, H-5′, CH_2_N); 3.29 (m, 2H, N-CH_2_); 3.06 (m, 8H, 2 × CH_3_ of N(CH_3_)_2_, CH_2_); 2.08, 2.05, 2.04 (3s, 9H, 3 × OAc); 1.72 (bs, 2H, CH_2_); 1.28 (bd, 8H, 4 × CH_2_); 0.82 (t, 3H, CH_3_); ^13^C NMR (125 MHz, D_2_O): 172.9, 172.6 (3C, 3 × OOCCH_3_), 99.5 (C-1), 71.0 (C-3), 70.6 (C-2), 68.5 (C-4), 65.3 (N-CH_2_), 62.9 (O-CH_2_), 62.8 (CH_2_-N), 61.1 (C-5), 58.0 (CH_2_), 51.5 (2C, N(CH_3_)_2_), 31.0, 28.2, 28.1, 25.6, 22.1, 21.9 (6 x CH_2_), 20.3, 20.2, 20.1 (3C, COOCH_3_), 13.4 (CH_3_); MALDI TOF-MS (CHCA): *m*/*z* 460.3 ([M-Br]^+^); HRMS (ESI): *m*/*z* ([M-Br]^+^) calculated for C_23_H_42_O_8_N: 460.2905; found: 460.2898.

***N*-[2-(2′,3′,4′-tri-O-acetyl-α-d-xylopyranosyloxy)ethyl]pyridinium bromide (5a)**—A reaction between compound **3** (37 mg, 0.094 mmol) and dry pyridine (0.5 mL) carried out for 24 h yielded compound 5a (33 mg, 75% yield, colorless oil); [$\alpha ]\begin{array}{c} 20\\ D \end{array}\;$+46.9° (c 1, MeOH); ^1^H NMR (500 MHz, D_2_O): δ 8.92, 8.60, 8.10 (d, t, t, 5 H, Py, J 5.9, J 7.7, J 7.7); 5.21 (t, 1H, H-3, J_3,4_ 9.4 Hz); 4.99 (bs, 1H, H-1); 4.96 (m, 1H, H-4); 4.89 (m, 1H, H-2, J_2,3_ 9.4 Hz); 4.89 (m, 2H, CH_2_-N, J 4.1 Hz); 4.22 (m, 1H, O-CH_2a_, J 11.1); 3.94 (m, 1H, O-CH_2b_ J 5.2); 3.78 (dd, 1H, H-5, J_4,5_ 5.6 Hz, J_5,5′_ 11.5 Hz); 3.24 (t, 1H, H-5′, J_4,5′_ 10.9 Hz); 2.03, 2.01, 1.90 (3s, 9H, 3 × OAc); ^13^C NMR (125 MHz, D_2_O): 173.4, 172.7 (3C, 3 × OOCCH_3_), 146.4, 145.1, 128.3 (5C, Py), 95.5 (C-1), 70.0 (C-2), 69.9 (C-3), 68.4 (C-4), 66.0 (O-CH_2_), 61.1 (CH_2_-N), 58.6 (C-5), 20.2, 20.1, (3C, COOCH_3_); MALDI TOF-MS (CHCA): *m*/*z* 382.2 ([M-Br]^+^); HRMS (ESI): *m*/*z* ([M-Br]^+^) calculated for C_18_H_24_O_8_N: 382.1496; found: 382.1492.

***N*-[2-(2′,3′,4′-tri-O-acetyl-α-d-xylopyranosyloxy)ethyl]-*N*,*N*,*N*-trimethylammonium bromide (5b)**—A reaction between compound **3** (30 mg, 0.079 mmol) and 33% trimethylamine in ethanol (0.5 mL) ended after 3 h. After removal of all volatile compounds, the residue was O-acetylated with Ac_2_O (0.5 mL) and pyridine (0.5 mL) at room temperature for 24 h. Purification of the crude product, as described in the general procedure, led to compound 5b (19.3 mg, 55%, yellowish oil); [$\alpha ]\begin{array}{c} 20\\ D \end{array}\;$+60.3° (c 1, MeOH); ^1^H NMR (500 MHz, D_2_O): δ 5.32 (t, 1H, H-3, J_3,4_ 9.3 Hz); 5.12 (d, 1H, H-1, J_1,2_ 2.8 Hz); 5.04 (m, 2H, H-2 and H-4, J_2,3_ 9.2 Hz); 4.17 (m, 1H, O-CH_2a_); 3.98 (m, 1H, O-CH_2b_); 3.92 (dd, 1H, H-5, J_4,5_ 5.6 Hz); 3.70 (t, 1H, H-5′, J_5′,5_ 11.5 Hz); 3.62 (m, 2H, CH_2_-N); 3.19 (s, 9H, 3 × CH_3_ of N(CH_3_)_3_); 2.08, 2.04, 2.01 (3s, 9H, 3 × OAc); ^13^C NMR (125 MHz, D_2_O): 173.4, 172.8, 172.5 (3C, 3 × OOCCH_3_), 96.0 (C-1), 70.2 (C-3), 69.9 (C-2), 68.6 (C-4), 65.6 (CH_2_-N), 62.0 (O-CH_2_), 58.9 (C-5), 54.1 (3C, N(CH_3_)_3_), 20.2, 20.1, 20.0 (3C, COOCH_3_); MALDI TOF-MS (CHCA): *m*/*z* 362.3 ([M-Br]^+^); HRMS (ESI): *m*/*z* ([M-Br]^+^) calculated for C_18_H_28_O_8_N: 362.1809; found: 362.1804.

***N*-[2-(2′,3′,4′-tri-O-acetyl-α-d-xylopyranosyloxy)ethyl]-*N*,*N*-dimethyl-*N*-hexylammonium bromide (5c)**—A reaction between compound **3** (35 mg, 0.092 mmol) and N,N-dimethylhexylamine (0.2 mL, 1.15 mmol) in acetonitrile (1.0 mL) carried out for 24 h yielded compound 5c (32 mg, 68%, colorless oil); [$\alpha ]\begin{array}{c} 20\\ D \end{array}\;$+49.9° (c 1, MeOH); ^1^H NMR (500 MHz, D_2_O): δ 5.29 (t, 1H, H-3, J_3,4_ 9.3 Hz); 5.11 (d, 1H, H-1, J_1,2_ 3.3 Hz); 5.06 (m, 2H, H-2, H-4); 4.15 (dd, 1H, O-CH_2a_); 3.93–3.91 (m, 2H, H-5, O-CH_2b_); 3.69–3.57 (m, 3H, H-5′, CH_2_N,); 3.37 (m, 2H, N-CH_2_); 3.12 (s, 6H, 2 × CH_3_ of N(CH_3_)_2_); 2.08, 2.05, 2.03 (3s, 9H, 3 × OAc); 1.77 (m, 2H, CH_2_); 1.31 and 1.28 (2 x bs, 6H, 3 × CH_2_); 0.81 (bs, 3H, CH_3_); ^13^C NMR (125 MHz, D_2_O): 173.4, 172.8, 172.5 (3C, 3 × OOCCH_3_), 96.1 (C-1), 70.3 (C-3), 69.8 (C-2), 68.8 (C-4), 65.6 (N-CH_2_), 62.3 (CH_2_-N), 62.0 (O-CH_2_), 58.7 (C-5), 51.5 (2C, N(CH_3_)_2_), 30.6, 25.3, 22.1, 21.8 (4C, 4 × CH_2_), 20.1 (3C, COOCH_3_), 13.2 (CH_3_); MALDI TOF-MS (CHCA): *m*/*z* 432.4 ([M-Br]^+^); HRMS (ESI): *m*/*z* ([M-Br]^+^) calculated for C_21_H_38_O_8_N: 432.2592; found: 432.2587.

***N*-[2-(2′,3′,4′-tri-O-acetyl-α-d-xylopyranosyloxy)ethyl]-*N*,*N*-dimethyl-*N*-octylammonium bromide (5d)**—A reaction between compound **3** (29 mg, 0.075 mmol) and N,N-dimethyloctylamine (0.2 mL, 0.972 mmol) in acetonitrile (1.0 mL) carried out for 24 h yielded compound 5d (28.5 mg, 70%, yellowish oil); [$\alpha ]\begin{array}{c} 20\\ D \end{array}\;$+44.7° (c 1, MeOH); ^1^H NMR (500 MHz, D_2_O): δ 5.29 (t, 1H, H-3, J_3,4_ 9.5 Hz); 5.11 (d, 1H, H-1, J_1,2_ 3.2 Hz); 5.07–5.04 (m, 2H, H-2, H-4); 4.15 (m, 1H, O-CH_2a_); 3.96–3.90 (m, 2H, H-5, O-CH_2b_, J_4,5_ 5.9 Hz); 3.71 (m, 1H, H-5′, J_5,5′_ 11.0 Hz); 3.56 (m, 2H, CH_2_N); 3.44 (dd, 2H, NCH_2_); 3.12 (s, 6H, 2 × CH_3_ of N(CH_3_)_2_); 2.08, 2.04, 2.03 (3s, 9H, 3 × OAc); 1.79 (m, 2H, CH_2_); 1.34–1.23 (m, 10H, 5 x CH_2_); 0.82 (t, 3H, CH_3_, J 6.6 Hz); ^13^C NMR (125 MHz, D_2_O): 173.2, 172.8, 172.5 (3C, 3 × OOCCH_3_), 96.1 (C-1), 70.4 (C-3), 69.9 (C-2), 68.8 (C-4), 65.6 (1C, NCH_2_), 62.2 (1C, CH_2_N), 62.1 (1C, O-CH_2_), 58.7 (C-5), 51.7 (2C, N(CH_3_)_2_), 31.0, 28.3, 28.1, 25.6, 22.3, 22.0 (6C, 6 × CH_2_), 20.3, 20.2, 20.1 (3C, COOCH_3_), 13.4 (CH_3_); MALDI TOF-MS (CHCA): *m*/*z* 460.3 ([M-Br]^+^); HRMS (ESI): *m*/*z* ([M-Br]^+^) calculated for C_23_H_42_O_8_N: 460.2905; found: 460.2899.

## Data Availability

Not applicable.

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
