# Peer review of "Synthesis, Antimicrobial and Mutagenic Activity of a New Class of d-Xylopyranosides"

_antibiotics, 2023, doi:10.3390/antibiotics12050888_

Round 1

Reviewer 1 Report

I have evaluated the manuscript (Antibiotics-23079650) titled “Synthesis, antimicrobial and mutagenic activity of a new class of D-xylopyranosides” by Dmochowska and coworkers. I found this article interesting for the readers and followed the journal Antibiotics’ scope. Overall presentation and discussion of this manuscript should be revised to make this article more interesting for the readers and better presentation of the work. Most of the discussion of chemistry should move to the supplementary.

I would recommend this article be published in Antibiotics after minor corrections. 

The author needs to address the following comments/corrections.

 1.     Supplementary should contain table of content.

2.     The introduction should be concise, and figures 1 to 8 should be consolidated into one figure.

3.     The author should discuss importance of present work in the introduction instead of more focus on previous work.

4.     The structure elucidation in the result and discussion should be concise, and bulk of this discussion should move to supplementary.

5.     The author should move the bulk of experimental part to the supplementary.

6.      Provide the reference for the reactions in scheme 1.

7.      The author should provide the reaction conditions in the arrow or in the footnote for scheme 1.

8.      All the NMR spectra should include peak picking. The COSY and HSQC spectra should be included in the manuscript.

9.      As the yield is poor for the glycosylation reaction, any explanation for this poor yield or any by product formation was observed. If the author loses the product during purification, then author should have in solution yield by Qnmr.

10.  Was the DFT calculation done to study the formation of both anomers for the glycosylation reaction?

11.  All the optimization of reaction condition for all reactions should be included in the supplementary in the tabular format.

12.  The author should mention the control (positive or negative) used for antimicrobial activity in the table.

13.  Any solubility study was performed for the final salts, and toxicity study?

14.  1H or 13C NMR (CDCl3, xx MHz) should contain the MHz of the NMR instrument.

Long sentences.

Reviewer 2 Report

In this manuscript, the authors have synthesized new group of D-xylopyranosides with an aglycone possessing a quaternary ammonium salt. All compounds were subjected for antibacterial as well anticancer studies where some of derivatives possess potential antibiotic properties against fungi (Candida albicans, Candida glabrata) and bacteria (Staphylococcus aureus, Escherichia coli) and weak mutagenic activities. 

My comments are as follows:

1. The scope and importance of this manuscript are high since a new type of potential derivatives of D- xylose were reported having good antibiotic properties with less mutagenic properties and might possess surfactant like properties. The route of the synthesis of the derivatives are also very feasible.

2. The manuscript is presented with rational scientific languages. The abstract and the introduction section represent well about the objective of this work.

3. Standard and authentic protocols was used to synthesize the derivatives 4a-d and 5a-d.

The molecules are characterized properly using NMR other spectroscopic techniques and seems to be pure.

Please integrate the 1H NMR spectrum of all derivatives.

Please incorporate COSY and HSQC spectrum as described in the manuscript.

The HPLC purities of all derivatives may be incorporated.

4. A standard protocol was used for in antibacterial studies. The minimum inhibition data showed that, the derivatives 4d and 5d perform better than the other derivatives and this shows the potential value of the derivatives. A comparative study could be performed with respect to a reference drug and for better understanding about the mode of action, molecular docking studies might be incorporated. The surfactant properties of the derivatives could be studied.

5. It is suggested to cite the following articles in the introduction section regarding importance of sugar derivatives with conformationally mobile glycosides and xylopyranosyl-paromomycin compounds in biological activity.

a) Daniil Ahiadorme, Chennaiah Ande, Rafael Fernandez-Botran, David Crich. Synthesis and evaluation of 1,5-dithialaminaribiose and -triose tetravalent constructs. Carbohydrate Research 525 (2023) 108781. (https://doi.org/10.1016/j.carres.2023.108781)

b) Rukshana Mohamad-Ramshan, Chennaiah Ande, Takahiko Matsushita, Klara Haldimann, Andrea Vasella, Sven N. Hobbie, David Crich. Synthesis of 4-O-(4-amino-4-deoxy-b-D-xylopyranosyl)paromomycin and 4-S-(b-D-xylopyranosyl)-4-deoxy-4’-thio-paromomycin and evaluation of their antiribosomal and antibacterial activity. Tetrahedron 135 (2023) 133330. (https://doi.org/10.1016/j.tet.2023.133330)

Reviewer 3 Report

The authors described the synthesis of 8 non-mutagenic D-xylopyranosides aglycone which exhibited antimicrobial activity against certain fungi and bacteria strains. Scientifically, I believe the manuscript is sound, however, the introduction section needs thorough editing to make it more concise, impactful, and enjoyable to read, e.g:

1.    "Taking into account the ease of obtaining from natural sources and the biological 27 activity of its derivatives, D-xylose, next to D-glucose, should probably be considered the 28 most important monosaccharide. Due to the fact that it can be obtained by hydrolysis of 29 hemicellulose, constituting up to 30% of this sugar [1,2], D-xylose is the second, after D- 30 glucose, most common monosaccharide found in nature."

The above sentences can be summarized into 1 or 2 sentences. They are providing the same information repeatedly. 

2.    Figures 1,2,3,4 can be combined into 1 panel. Figures 5 and 6 can be combined. 

3.     For the purification of anomers, have the authors looked at SFC method separation? These columns are also useful for diastereomers separation. 

4.     The authors should justify their choice of pyridine and the other three tertiary amines. The subset of selected amines is quite limited. In fact, the alpha anomer was an undesired isomer of their glycosidation reaction. 

5.     It is interesting that 5d shows better MIC than 4d, authors should speculate on the result and discuss it in the summary section

Minor concerns:

Please report the 13C resonances to one decimal point

There is no significance in reporting rf =0.0 if the compounds are not moving in the solvent system chosen

In Experimentals, there are several places that needed proof-reading. 

The English language needs minor modification. 

Round 2

Reviewer 2 Report

The authors have addressed all my comments for this paper and answered the technical questions I have for this article. The paper has been significantly improved after revising so and I recommend publication in antibiotics.